# Crystalline polymeric carbon dioxide stable at megabar pressures

Kamil F. Dziubek [1], Martin Ende[2], Demetrio Scelta [1,3], Roberto Bini[1,3,4], Mohamed Mezouar [5], Gaston Garbarino[5] & Ronald Miletich[2]

Carbon dioxide is a widespread simple molecule in the Universe. In spite of its simplicity it has a very complex phase diagram, forming both amorphous and crystalline extended phases above 40 GPa. The stability range and nature of these phases are still debated, especially in view of their possible role within the deep carbon cycle. Here, we report static synchrotron X-ray diffraction and Raman high-pressure experiments in the megabar range providing evidence for the stability of the polymeric phase V at pressure-temperature conditions relevant to the Earth's lowermost mantle. The equation of state has been extended to 120 GPa and, contrary to earlier experimental findings, neither dissociation into diamond and $\varepsilon$-oxygen nor ionization was observed. Severe deviatoric stress and lattice deformation along with preferred orientation are removed on progressive annealing, thus suggesting $CO_2$-V as the stable structure also above one megabar.

[1] LENS, European Laboratory for Non-linear Spectroscopy, Via N. Carrara 1, I-50019 Sesto Fiorentino, Firenze, Italy. [2] Institut für Mineralogie und Kristallographie, Universität Wien, Althanstrasse 14, A-1090 Wien, Austria. [3] ICCOM-CNR, Institute of Chemistry of OrganoMetallic Compounds, National Research Council of Italy, Via Madonna del Piano 10, I-50019 Sesto Fiorentino, Firenze, Italy. [4] Dipartimento di Chimica "Ugo Schiff" dell'Università degli Studi di Firenze, Via della Lastruccia 3, I-50019 Sesto Fiorentino, Firenze, Italy. [5] European Synchrotron Radiation Facility, 71 avenue des Martyrs, CS 40220, 38043 Grenoble Cedex 9, France. Correspondence and requests for materials should be addressed to K.F.D. (email: dziubek@lens.unifi.it)

Carbon dioxide ($CO_2$) is an important compound not only for chemists and physicists, but also in the planetary sciences. $CO_2$ is a terrestrial volatile compound, being one of the most important carbon species in the Earth's mantle[1]. The occurrence of $CO_2$ and its speciation is pivotal to understanding transport mechanisms and geochemical reservoirs within the deep carbon cycle and its influence on major geodynamical processes, including melting of mantle rocks, the formation of carbonates, and release into the atmosphere[2–6]. The structural incompatibility with solid silicates and their melts give rise to debates upon the occurrence and role of free $CO_2$ under deep mantle conditions[1], in particular with respect to the origin of mantle plumes and the formation of superdeep diamonds at the lowermost mantle close to the core-mantle boundary[2–9].

Despite its simple stoichiometry, $CO_2$ exhibits complex phase behavior under non-ambient conditions. Below 40 GPa it was reported to exist in the form of five different crystalline polymorphs, all of them made up of isolated $CO_2$ molecules. Detailed reviews on the structural and phase complexity of the crystallographically different $CO_2$ phases can be found elsewhere[10–12]. On further compression, the unsaturated molecular species transform into a polymerized non-molecular network structure characterized by the increased coordination number of carbon. This is exemplified in amorphous carbonia glass containing carbon in three-fold and four-fold coordination[13,14], which forms at pressures above 40 GPa and considerably moderate temperatures (up to 680 K). In contrast, several laser heating experiments between 1500 and 3000 K[15-18] give evidence for the crystalline phase $CO_2$-V to occur within the 40–50 GPa range. Its crystal structure was determined and found to resemble that of tetragonal $\beta$-cristobalite, space group $I\bar{4}2d$, consisting of a three-dimensional network of corner-sharing $CO_4$ tetrahedra[17,18]. To the best of our knowledge, only in one study the sample of $CO_2$-V was obtained at higher pressures (41–99 GPa) and temperatures ranging up to 2000 K[19]. However, the authors reported exclusively the cell volume variation. Nevertheless, the thermodynamic and structural stability and the suggested variability of the phase V structure apparently depend on the sample history and on the pressure-temperature pathway, and are even currently stirring controversies[20–22].

Here, we report the results of X-ray diffraction (XRD) and Raman scattering experiments on a $CO_2$ sample synthesized at conditions exceeding the megabar (120 GPa) and temperatures typical of the lower mantle (2700 K). The analysis reveals that the laser heating experiment yields pure polymeric phase V, demonstrating that the other forms of $CO_2$ suggested to exist in this pressure-temperature window are actually metastable. Moreover, we do not find any proof for dissociation of $CO_2$ into carbon and oxygen. Our results indicate that the crystalline extended form of $CO_2$ is stable in the thermodynamic conditions of the deep lower mantle, and therefore could be helpful to understand the distribution and transport of carbon in the depths of our planet.

## Results

**Sample preparation and XRD characterization.** A sample of $CO_2$ was loaded into a diamond anvil cell (DAC) together with some specks of cryptocrystalline magnesite ($MgCO_3$) and compressed at room temperature to about 85 GPa while monitoring the XRD pattern. In agreement with the earlier reports[13,14], a progressive broadening and full disappearance of the Bragg peaks evidenced the pressure induced amorphization (Supplementary Fig. 1). However, it needs to be emphasized that amorphization under cold compression leads to a kinetically trapped metastable form, which cannot be defined as a thermodynamic energy-

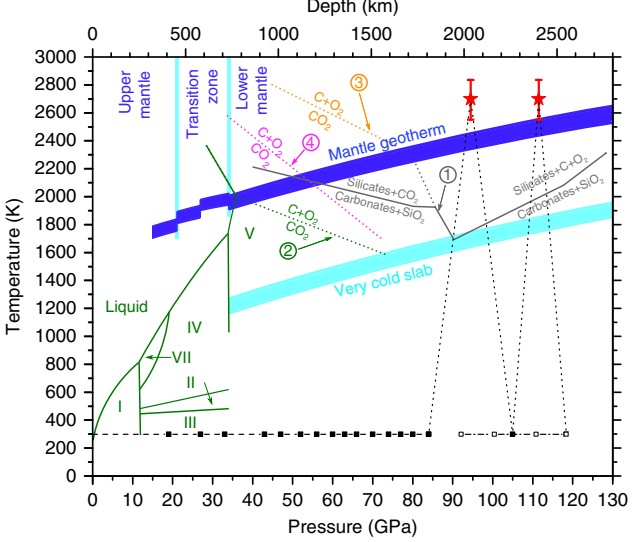

**Fig. 1** Experimental pressure-temperature pathway and the phase diagram of $CO_2$. The plot shows the relevant parts of the phase diagram of $CO_2$ and the $CO_2$ breakdown lines reported in previous studies. 1 – Phase boundaries in the $MgCO_3$-$SiO_2$ system after ref.[9] (gray lines); the dotted gray line corresponds to the breakdown of $CO_2$. 2 – Phase relations and melting curves (solid green lines) and $CO_2$ breakdown line (dotted green line) after ref.[8] and references therein; phase boundaries of $CO_2$-II and $CO_2$-III after ref. 12. 3 – Breakdown reaction after ref.[27] (dotted brown line). 4 – Breakdown reaction after ref.[28] (dotted purple line). The dark blue line corresponds to the adiabatic temperature profile in the mantle geotherm after ref.[50], the light blue line to the depth-temperature path in ref.[9] as referred to a very cold subduction slab. The dashed black line follows isothermal compression up to ~ 85 GPa, the dotted black lines show the putative heating and cooling route, the dash-dot line represents decompression run. Squares denote experimental points at which XRD patterns were measured (filled on compression, empty on decompression). Red asterisks mark P-T conditions at laser heating cycles with the error bars at the estimated uncertainty of ± 150 K, as explained in the Methods section

minimum state at given conditions. Upon subsequent laser heating for 5–10 min to ~2700 K (Fig. 1) the amorphous material transformed into a polycrystalline product. $MgCO_3$ served as a sufficient absorber of the laser energy, with the $\nu_2$ in-plane bending and $\nu_4$ out-of-plane bending as the active infrared modes of the $CO_3^{2-}$ anion[23] coupling well with the $CO_2$ laser (wavelength 10.6 μm corresponding to frequency 943 cm$^{-1}$). The XRD patterns collected after temperature quenching were indexed with a tetragonal cell of $CO_2$-V reported previously[17,18], while additional weak lines can be assigned to the rhenium (Re) gasket material (Fig. 2). The presence of the minor broad features flanking the either side of the $CO_2$-V 112 reflection was already reported in previous experiments and they were attributed either to the Re gasket[17], or to the untransformed molecular $CO_2$ phases[18]. The latter conclusion was supported by the benchmark experiment on a crystal of $CO_2$ embedded in helium (He), where all the diffraction lines in the XRD pattern of a material obtained by the laser heating at comparable pressure and temperature conditions could be attributed to $CO_2$-V phase, Re or He. In our measurements both these interpretations could be ruled out. During the initial compression of the molecular phases of $CO_2$ at room temperature we completely transformed them to amorphous material with XRD pattern lacking sharp diffraction peaks (Supplementary Fig. 1). As the reappearance of molecular phases during laser heating is highly unlikely and the weak but still distinguishable Re diffractions lines are observed elsewhere in the

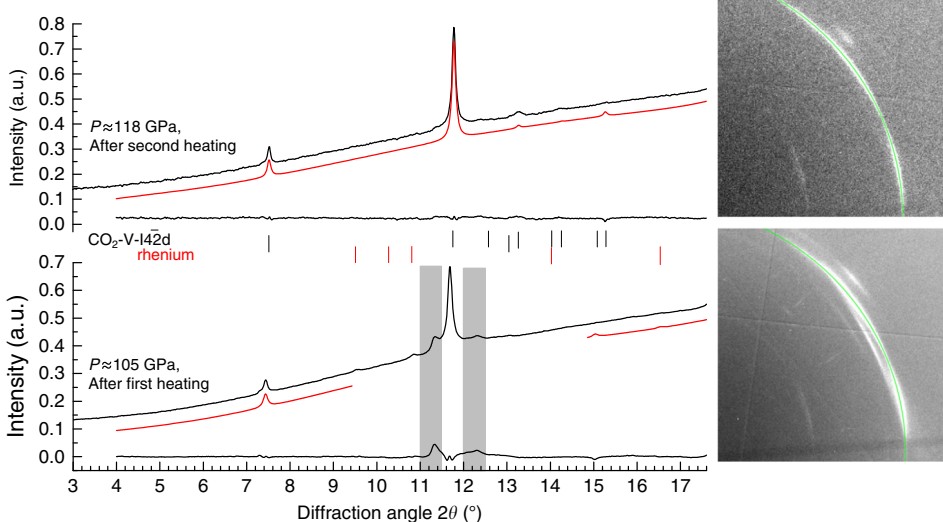

**Fig. 2** Integrated X-ray diffraction patterns. The data are collected after the first (bottom) and second (top) cycle of laser heating, with the temperature quenched to 295 K between the two cycles (Black lines—experimental data, red lines—calculated profiles; difference line at the bottom). The calculated peak positions for the $CO_2$-V phase and rhenium lines are displayed as tick marks. Additional weak features of intensities in the pattern (as indicated by the gray area) can be attributed to intracrystalline strain gradients and orientation-dependent lattice deformation of differently oriented crystal grains and completely disappear after the second cycle of heating. The corresponding 2D diffraction images are shown at the right (112 peak position indicated with green arcs)

pattern, we suggest that these additional features can be explained by the presence of differently oriented highly stressed domains in non-hydrostatic conditions. This supposition is corroborated by the occurrence of partial Debye-Scherrer rings showing diverse preferred orientation patterns for different domains (Fig. 2). After the second cycle of laser heating the stress conditions relaxed resulting in disappearance of extra weak peaks while the reflections of $CO_2$-V became sharper (FWHM of 112 reflection decreases by 25%) and less textured.

In order to examine thoroughly the homogeneity of the sample and to search for any potential products of chemical reaction between $CO_2$ and other materials inside the DAC (MgCO3, ruby, Re gasket) we scanned the sample chamber horizontally and vertically in the focal plane of the X-ray beam. While we identified diffraction peaks in the lateral regions of the sample corresponding to high-pressure high-temperature Rh2O3(II)-type phase of alumina, as remnants of a transformed ruby[24], and β-ReO2 Re(IV) oxide, the product of reaction between $CO_2$ and Re gasket[25] (Supplementary Fig. 2), the central part of the sample chamber was found to be free of any side- or by-products. In particular, we have not found any evidence of magnesite, magnesium oxide MgO, diamond, nor ε-oxygen with a fine mesh of 2 μm over the entire sample area.

**Equation of state and lattice parameters**. We subsequently released the pressure on the fully transformed sample in order to follow the variation of lattice parameters and volume with pressure. The change of the parameter $a$ and of the volume per chemical formula unit is in good agreement with previous experiments and DFT predictions (Fig. 3) and consistent with the previously found equation of state[18]. The slight difference with respect to earlier results can be explained by the non-hydrostatic effects in a stressed sample at megabar pressures, however it should be noted that our results are remarkably well described by calculations using GGA approximation[18,26]. We have therefore proposed alternative fit of the equation of state based on our results and literature data on $CO_2$-V embedded in He[18]. It could be refined to $B_0 = 114(5)$ GPa and $B_0' = 5.7(3)$ while fixing the $V_0/Z$ at 22.75 Å. On the other hand, it should be emphasized that

the $c$ parameter is noticeably longer than the previously observed experimental values up to 65 GPa. It agrees with the reported calculations, which indicated the possible negative linear compressibility of $c$ parameter in the range in consideration. We have noticed, however, that for our sample the $c$ parameter, as well as the $a$ parameter, increase while pressure is decreased. This raise a question if the observed trend for $c$ could be attributed to the presence of a maximum in the pressure dependence between 65 and 92 GPa.

**Raman scattering measurements**. Complementary to the XRD analysis we have collected Raman spectra at the pressure of about 110 GPa (Fig. 4). All the peaks were assigned to the modes of $CO_2$-V reported elsewhere for experimental bands[12] and calculated values[17] by linear or slightly non-linear extrapolation of these low pressure data also providing an excellent agreement as the intensity ratio is concerned. The excellent quality of these spectra should be also remarked as, for example, by the observation of the perfectly resolved doublet at 600 cm$^{-1}$.

**Discussion**

This study is to our knowledge the first experimental work on $CO_2$ after annealing at conditions close to the core-mantle boundary and at a pressure and temperature regime relevant to the geotherm. Previous studies on $CO_2$ in extreme pressure and temperature range suggested the decomposition of $CO_2$ with the formation of diamond and ε-oxygen[8,27,28]. Their proposed reaction boundary has a negative pressure-temperature slope, indicating that above 100 GPa $CO_2$ should dissociate even below 2000 K, therefore indicating that the $CO_2$-V phase can be stable only near the top of the lower mantle and dissociates at greater depths. Moreover, another study claims that upon laser heating to ~1700–1800 K at 85 GPa, $CO_2$ transforms into a new extended ionic solid form, and also for this transition negative pressure-temperature slope was proposed[29]. On the contrary, the remarkable stability of $CO_2$ and no evidence for decomposition up to at least 200 GPa and 10000 K was found by recent calculations[30,31]. While the results of the experiment may seem

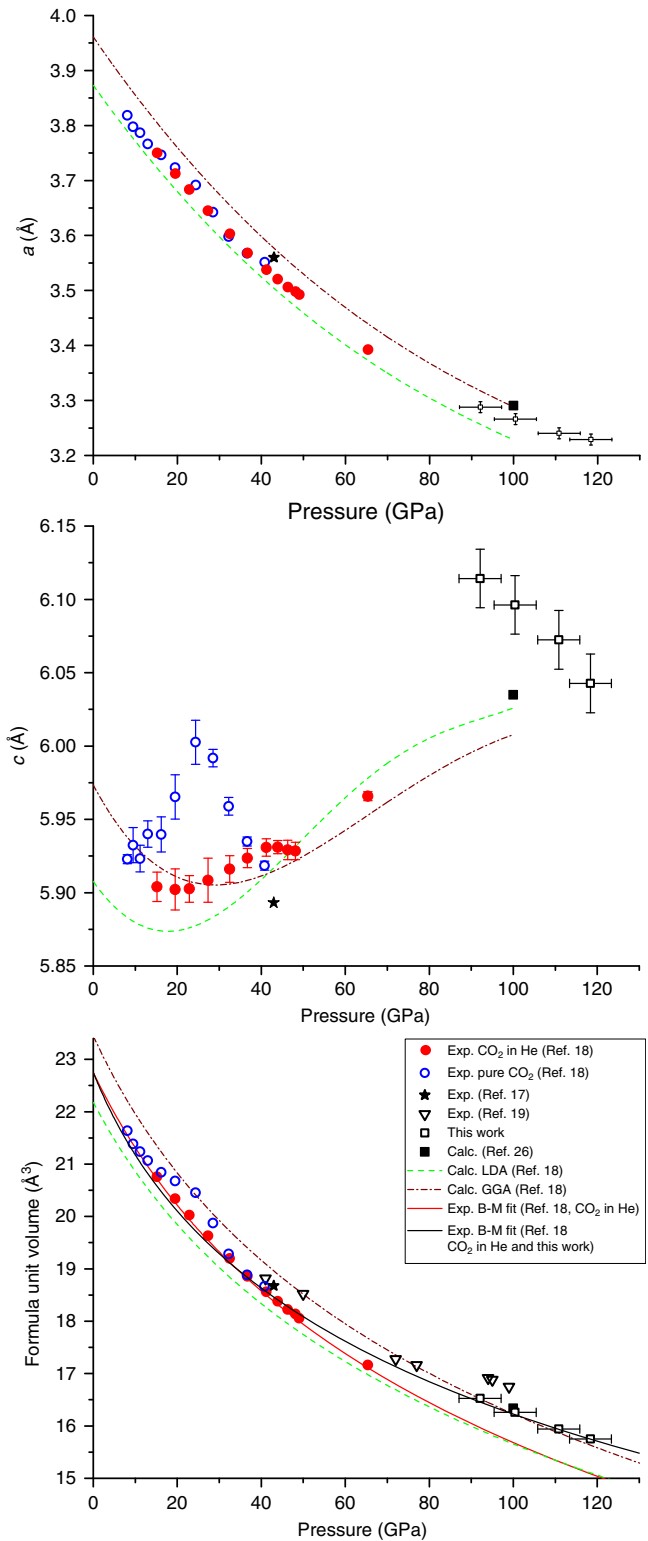

**Fig. 3** Pressure dependence of lattice parameters and formula unit volume. The data for $CO_2$-V phase are compared to the literature experimental (scattered points) and theory data (dashed lines, cf. the inset). The fitted equations of states are indicated by solid lines. The experimental points determined in this work are presented with the bidirectional error bars, representing the uncertainties of lattice parameters and volume determined as obtained from the profile fitting, and ± 5 GPa uncertainty of pressure determination (see Methods section)

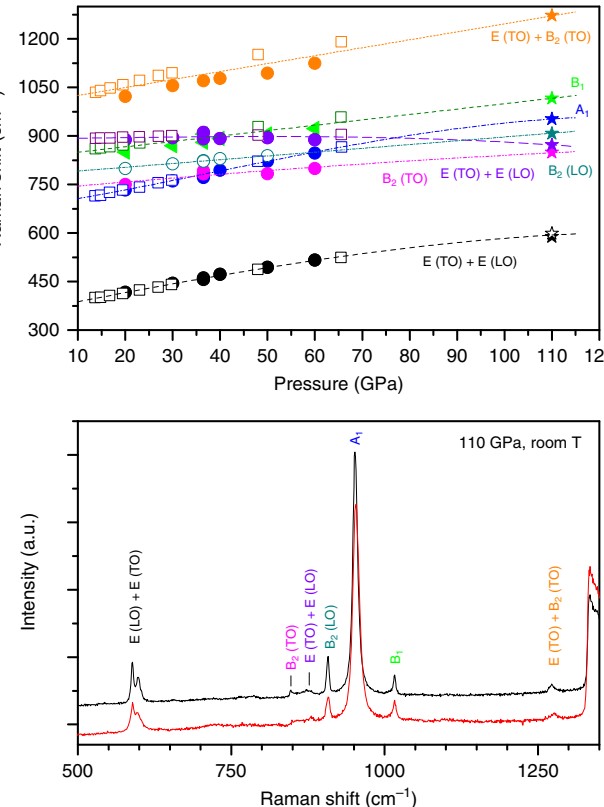

**Fig. 4** Raman data. The data sets are acquired in two different points of the sample at ~110 GPa after quenching to room temperature with the mode assignment (bottom panel). Pressure dependence of the frequencies of Raman modes (top panel). Empty symbols—experimental data[12], full symbols—calculations[17], asterisks—this study

controversial, dissociation into carbon and oxygen has been reported in laser heating experiments at pressures not exceeding 70 GPa and temperatures of the order of 1800–2800 K[8,27,28]. Our experiments have been performed at higher pressures thus falling into the area of dissociation, according to a linear extrapolation of the reported equilibrium boundary with a pronounced negative slope (~11.5 K/GPa). However, other authors did not observe dissociation of $CO_2$ heated above 1800 K at 85 GPa at all[29]. As the occurrence of $\beta$-$ReO_2$ is spatially restricted to the lateral regions inside the sample chamber in our study, it suggests that the breakdown of $CO_2$ most likely originates from a reaction with the Re gasket. This in turn would provide a plausible explanation for the reported discrepancies on the $CO_2$ decomposition[8,9]. Another interpretation of this discrepancy can be a substantial difference in pressure between the maximum attained value in the literature data (~70 GPa) and our results. It is widely known that compression strongly disadvantages an increase of molecularity and thus the dissociation of $CO_2$. Therefore, the breakdown lines reported in previous works may not linearly extrapolate to higher pressure. In fact, it is possible that in the studied range dissociation requires considerably higher temperature than 2700 K.

Our results are perfectly in line with these findings thus evidencing that $CO_2$-V can be stable in the absence of chemical reactions with other elements at P,T conditions corresponding to depths of 2000–2400 km in the deep lower mantle (cf. Figure 1). The stability issue of extended $CO_2$ phases and their breakdown is of utmost geochemical importance with reference to the fact that the dissociation would produce diamond and oxygen, which in

turn increases the oxygen fugacity by several orders of magnitude and affects the formation of superdeep diamonds[8,9,27], the questionable existence of highly oxidized dense iron oxides[32–36], and possible significant conductivity anomalies related to the changed redox state in lower-mantle minerals[8]. There is mutual agreement about carbonate decomposition and $CO_2$ release by reaction with silicate minerals at pressures above ∼55 GPa following the mantle geotherm[8,37], in contrast to opposing findings on the subsequent breakdown reaction. The presence of $CO_2$ extended phases at lower-mantle conditions could represent an important carbon source for reactions with oxide minerals (e.g., magnesiowüstite) to form iron-rich carbonates with tetrahedrally coordinated carbon[38,39] and to allow transport of carbon down to the core-mantle boundary, or to get reduced to immobile diamonds due to the lower-mantle reducing conditions and be at the origin of deep melting upon upwelling of mantle rocks[40].

## Methods

**Sample preparation**. $CO_2$ gas (99.99% purity, Linde) and natural anhydrous cryptocrystalline gel magnesite ($MgCO_3$) were used for DAC loading. All measurements were carried out in a symmetric membrane DAC[41] equipped with bevelled diamond anvils (150 μm culet). Re foils with an initial thickness of 200 μm, pre-indented to 30 μm, were used as the gasket material. The laser drilled sample chamber of 50 μm diameter contracted in the course of the experiment to ca. 40 μm. Liquefied $CO_2$ was loaded cryogenically at −25 °C and 23 bar into the DAC, together with a tiny ruby sphere for controlling the initial pressure and several chips of $MgCO_3$, which facilitated heating on the sample due to its laser absorbance. No thermal insulation of the diamonds was applied.

**XRD measurements**. Angle-dispersive XRD patterns were collected at ID27 beamline of the ESRF with a high brilliance synchrotron radiation from a two phased undulator set to 33.3 keV (wavelength of 0.3738 Å) and focused down with Kirkpatrick–Baez mirrors to about $3 \times 4$ μm FWHM. XRD patterns were collected using a MAR 165 CCD detector placed about 245 mm from the sample. The DAC was rotated by ± 7° during the exposure and a typical data acquisition time was 15 s. High-temperature conditions were achieved at the experiment using the online $CO_2$ laser heating system (wavelength 10.6 μm), in two cycles of 5–10 min each. After each heating cycle and subsequent temperature quenching the pressure increased spontaneously while keeping the DAC membrane pressure constant, first from 90 to 110 GPa and then to 120 GPa. Pressures were monitored from diffraction of the gasket edge and applying the thermal equation of state of Re[42]. The uncertainty of pressure determination was estimated as ± 5%, i.e. about ± 5 GPa at megabar conditions. The temperature was measured basing on the sample thermal emission (black body radiation) collected using Schwarzschild mirrors[43]. Considering various sources of error, including both radial and axial temperature gradients in the laser-heated sample, pressure evolution of the sample emissivity and temperature fluctuations with time, the total uncertainty of the temperature determined using the Planck radiation function can be estimated to be around ± 150 K for the experimental setup employed in this study[44].

**Raman scattering measurements**. The experiments were carried out using the 647.1 nm line of a Kr+ laser as the excitation source. The emission was collected at backscattering geometry employing a Mitutoyo 20 × long working distance microobjective, dispersed by an Acton/SpectraPro 2500i single monochromator with a 900 groove mm$^{-1}$ grating, and followed by detection using a CCD (Princeton Instruments Spec-10:100 BR)[45]. The spectral resolution of the measurements was 0.7 cm$^{-1}$ and transverse spatial resolution about 3 μm.

**Analysis of experimental data**. The collected 2D images were radially integrated using the Dioptas software[46]. Analysis of the XRD data was carried out using the TOPAS software[47]. The Raman spectra were analyzed using the Fityk program[48], fitting the bands by pseudo-Voigt profiles. Equation of state was fitted using EosFit7-GUI[49].

**Data availability**. Further details of the crystal structure investigations may be obtained free of charge from FIZ Karlsruhe, Germany through the hyperlink: https://www.fiz-karlsruhe.de/en/leistungen/kristallographie/kristallstrukturdepot/order-form-request-for-deposited-data.html, on quoting the deposition numbers CSD-434505 and CSD-434506.

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

## Acknowledgements

This work was supported by the Deep Carbon Observatory initiative (Extreme Physics and Chemistry of Carbon: Forms, Transformations, and Movements in Planetary Interiors, from the Alfred P. Sloan Foundation). The X-ray diffraction measurements were carried out at the ID27 beamline, European Synchrotron Radiation Facility (ESRF), Grenoble, France. The authors thank the ESRF Sample Environment Support Service for provision of a loan pool diamond anvil cell and Dr. Harald Müller of the ESRF Chemistry Laboratory for his help with sample loading. We would like to acknowledge the anonymous reviewers for their valuable and constructive comments that greatly contributed to improving the manuscript.

## Author contributions

K.F.D. and R.M. conceived the study; K.F.D. and M.E. prepared the sample; K.F.D., M.E., D.S., M.M., and G.G. conducted the experiments, K.F.D., M.E., and D.S. processed the experimental data; K.F.D., M.E., D.S., R.B., and R.M. analysed and interpreted the data; and K.F.D. wrote the paper with contributions from all other authors who commented on drafts and approved the final version of the manuscript.

## Additional information

**Competing interests:** The authors declare no competing interests.

