## [Peer Review File · Nature Communications]

Reviewer #1 (Remarks to the Author):

see attachment

Reviewer #2 (Remarks to the Author):

Dzuibek and co-author report new experimental data on carbon dioxide (CO₂) at extreme conditions. The authors performed laser-heated diamond anvil cells experiments with in situ X-Ray Diffraction as well as Raman spectroscopy characterization on CO₂ loaded together with cryptocrystalline gel magnesite. To my knowledge, this is the first study performed at pressure above the megabar and at temperature that are relevant for the Earth's lower mantle. The main outcome is the experimental evidence of the stability of polymeric phase CO₂-V at deep lower mantle conditions. This results are in line with recent theoretical calculation but in contrary to all previous experimental studies which predicted dissociation of CO₂ at these conditions. Therefore, their results has potentially great significance not only in material science but also in earth and planetary science. I see no reason why this should not, with revisions, become an good publication.

However, one of my main concerns is the lack of characterization at high temperature. All data reported are from analyses at high pressure but room temperature after laser heating. How can the authors be confident that there has been no back transformation upon temperature quenching? This is particularly important as the authors seem to be willing to discuss the deep earth interior. I note, however, that in all previous studies data were also taken at room temperature after laser heating, so I assume it might come from a technical challenge.

Below I outline my specific comments:

Line 35 : I suggest adding reference 2 as the potential influence of CO₂ on the origin of mantle's plume was the main focus of Isshiki et al.(2004) discussion

Line 37 – 38: To my knowledge CO₂ phase IV is an isolated CO₂-bearing phase but not linear CO₂ (Yoo et al, PRL, 2001), therefore, below 40 GPa, all 5 phases are made of isolated CO₂ molecular group but not linears C=O=C molecules.

Extending Figure 1 : This figure is particularly difficult to “read” in its present stage, I would suggest plotting less spectra in order to make them bigger and labeling. What are the new XRD peaks that appears at 27 GPa : Re?

Line 65-69: I find these lines confusing as it does seem that the authors are suggesting that Re and molecular CO₂ are two possibilities to explain their data. It is only a few lines later that these are refuted.

Line 78: A figure with a “cake” of the image plate would be great to prove their theory on highly stressed domains (in sup. Material for example). In addition to the 112 peak; we do see a shoulder on the 101 XRD peak at 105 GPa but can not see it in the partial image plate as it is presented right now. I would expect laser heating to release any stress, how come after heating at 2700 K for a few minutes there is still stress? Was the diffraction taken in the heating spot?

Line 82: I am not convinced that the diffraction is “less structured” as we still see partial Debye Scherrer rings at 118 GPa.

Extended Figure 2: Where is the reflexion 110 of β -ReO₂? According to Figure 5 of reference 24, I would expect a diffraction peak at roughly $2\theta = 6$.

Also, Reference #24 should be added to the figure caption.

Discussion and Conclusion: In order to meet Nature Communications standards, I think more discussion on the implications of the results is needed. As the authors seem to be willing to discuss their results in terms of Earth Science, I would expect some discussion about the difference in having decomposition of CO₂ with release of oxygen at lower mantle conditions and the stability of a solid CO₂ phase in terms mantle dynamic and deep carbon cycle (see Isshiki et al 2004; and eventually Hu et al., Nature 2016; Yagi, Nature 2016; Boulard et al. JGR 2012)

Methods :

line 296, it is well know that the pressure measured into the Re gasket is significantly lower than the sample at the center of the sample chamber, is that taken into account in the present pressure measurement?

Extended Figure 3: What are the axis label? Am I understanding correctly that the XRD map was obtain by measuring Re unit cell volume? Does it mean Re diffraction peaks were measurable even in the center of the sample chamber?

As Re should only be present in the gasket but not within the sample chamber, I can not explain to myself why a pressure change would be observed within the sample chamber if only looking at Re XRD peaks.

Reviewer #1

1. Previous experiments clearly showed that CO₂ breaks down to form carbon and oxygen at high P-T conditions. The authors need to offer an explanation for why there is a difference. It could be that the present experiments were carried out at lower temperatures. Much more information needs to be presented on how they determined temperatures.

*In our study temperature was determined basing on the sample thermal emission using the Planck radiation with the uncertainty of 150 K. Thus more details about the technique were added in the 'methods' section of the manuscript. In previous studies pure CO₂ was loaded in a diamond anvil cell, while in our experiment chips of MgCO₃ were used as the IR-laser absorber. CO₃²⁻ anions of magnesite couple much better with the wavelength of CO₂ infrared laser than polymeric CO₂-V, which explains why indeed in our experiments we reached higher temperatures. Our experimental findings show that the dissociation of CO₂ can be justified regarding the possible reaction of CO₂ with the gasket material at the pressure-chamber wall. Indeed, Santamaría-Pérez et al., *Inorg. Chem.* 55, 10793-10799 (2016), have studied recently the reactivity between carbon dioxide and various transition metals including Au, Pt and Re in a laser-heated DAC. According to their findings there is no effect of gold on CO₂, platinum induces the decomposition into the elements, while rhenium is oxidized by CO₂ forming ReO₂. In all previous studies dissociation of CO₂ was reported in laser heating experiments at pressures not exceeding 70 GPa and temperatures of the order 1800-2000 K. Tschauner et al., *Phys. Rev. Lett.*, 87, 075701 (2001), did not mention explicitly the gasket material, but nevertheless stated in their article: "We note that the signal from O₂ is stronger outside rather than inside the heated area. This is consistent with the visual observation of an opaque ring surrounding the heated zone." Such an observation suggests that the decomposition reaction starts at the interface with the gasket, and continues proceeding to the center of the pressure chamber. This seems rather counterintuitive, since one can expect that the decomposition reaction gives a higher yield in the center of the sample, facilitated by higher temperature in place of direct heating, if actually promoted by thermodynamics. In our study we observed ReO₂ in the lateral part of the sample chamber (close to the gasket edge) while the central zone consisted only of CO₂-V. Moreover, the relative peak intensities of the CO₂ phase are indeed lower in the lateral zone, which encouraged us to assume a higher degree of CO₂ breakdown in this area thus confirming the observation reported by Tschauner et al. We presume that the relatively short time of laser heating was not sufficient for the reaction completely to progress towards the center. Therefore we are not surprised that Yoo, et al., *Angew. Chem. Int. Ed. Engl.* 50, 11219-11222 (2011) and Yoo, et al. *Phys. Chem. Chem. Phys.*, 15, 7949-7966 (2013) did not observe any dissociation of CO₂ during laser heating above 1800 K and 85 GPa. This agrees also well with the calculations by Boates et al. *Proc. Natl Acad. Sci. USA* 109, 14808-14812 (2012), who did not predict any decomposition of CO₂ up to at least 200 GPa and 10000 K. Finally, the papers by Tschauner et al. and Litasov et al. reported a boundary for dissociation of CO₂ with a steep negative slope (~11.5 K/GPa), which implies ambient temperature dissociation at about 180 GPa, which definitely has not been observed. Hence, we included the above mentioned explanation in the manuscript, reassessing critically previous reports and indicating the possibility of reaction between CO₂ and metallic gasket/laser coupler. We have also re-plotted the Fig.1, which is now even more relevant to geoscientific implications.*

2. Line 48: The authors comment on the work of Ref. 17. What that study did not report is irrelevant. The sentence containing 'cell volume variations' should end there.

We have shortened this sentence as suggested.

3. The next sentence states that phase boundaries are dependent on sample history. They are not. Most readers will interpret this to mean equilibrium phase boundaries, which are well defined.

We have removed a fragment “the phase boundaries” from this sentence, as suggested.

4. Line 59 comments on ‘carbonia’. The disappearance of signal cannot be taken as the indication of a transition. The authors need to show actual results (e.g., by spectroscopy) for the formation of a new phase. Having said this, the discussion of ‘carbonia’ is irrelevant to the geoscience implications of the paper.

We have not performed spectroscopic studies of the compressed CO₂ sample prior to laser heating, and we agreed therefore to replace in this part of the manuscript term ‘carbonia’ with ‘amorphous material’.

Reviewer #2

One of my main concerns is the lack of characterization at high temperature. All data reported are from analyses at high pressure but room temperature after laser heating. How can the authors be confident that there has been no back transformation upon temperature quenching? This is particularly important as the authors seem to be willing to discuss the deep earth interior. I note, however, that in all previous studies data were also taken at room temperature after laser heating, so I assume it might come from a technical challenge.

The reviewer is absolutely right that in situ characterization at high temperature is a real challenge, especially for a low-Z material studied in our work. Actually, the experiments in the laser-heated diamond anvil cells typically raise the opposite question. The very fast recovery to the ambient temperature can lead to quenching of phases stable at high temperature but metastable at room temperature. In our case temperature quenching was particularly fast because no insulation of the sample from the diamonds was adopted (to prevent potential reactions with the insulator material). It should be also emphasized that CO₂-V is an extended covalent structure, hence its transition to other crystal phase requires a substantial activation energy to overcome the kinetic barrier. Having said that, further in situ experiments at high pressure-temperature conditions would be of great interest indeed.

Line 35 : I suggest adding reference 2 as the potential influence of CO₂ on the origin of mantle’s plume was the main focus of Isshiki et al.(2004) discussion

We have added the suggested reference as well as other references related to Earth science.

Line 37 – 38: To my knowledge CO₂ phase IV is an isolated CO₂-bearing phase but not linear CO₂ (Yoo et al, PRL, 2001), therefore, below 40 GPa, all 5 phases are made of isolated CO₂ molecular group but not linears C=O=C molecules.

Actually, in all the molecular phases the O=C=O molecules are linear including phase IV. The data reported in the work mentioned by the reviewer were later confuted by different groups, see Gorelli et al.

Phys. Rev. Lett. 93, 205503 (2004) and Datchi et al. *Phys. Rev. Lett.* 103, 185701 (2009). However, this fact is irrelevant to the implications of our paper and we agree to omit the word 'linear' in the text.

Extending Figure 1 : This figure is particularly difficult to “read” in its present stage, I would suggest plotting less spectra in order to make them bigger and labeling. What are the new XRD peaks that appears at 27 GPa : Re?

We have reduced the number of patterns as requested. The peak around 9.1 deg at 27 GPa comes indeed from Re as its most intensive diffraction line and is present basically in all the patterns at various intensity (usually very low). We might have not taken all the XRD patterns at the same point of the sample as the dimensions of the pressure chamber changed with pressure.

Line 65-69: I find these lines confusing as it does seem that the authors are suggesting that Re and molecular CO₂ are two possibilities to explain their data. It is only a few lines later that these are refuted.

This part of the text do not refer to our experiments but to the data in papers by Santoro et al., Proc. Natl. Acad. Sci. USA 109, 5176-5179 (2012) and Datchi et al., Phys. Rev. Lett. 108, 125701 (2012). While they justified extra features in their XRD patterns by the presence of Re lines or untransformed molecular CO₂, in our case we concluded that neither of this explanations work. We suggested instead that the additional features can be explained by the presence of differently oriented domains of CO₂-V in non-hydrostatic conditions. We have clarified this issue and appropriately changed the wording in the relevant part of the text.

Line 78: A figure with a “cake” of the image plate would be great to prove their theory on highly stressed domains (in sup. Material for example). In addition to the 112 peak; we do see a shoulder on the 101 XRD peak at 105 GPa but can not see it in the partial image plate as it is presented right now. I would expect laser heating to release any stress, how come after heating at 2700 K for a few minutes there is still stress? Was the diffraction taken in the heating spot?

Below we attach “cakes” of the CCD images, based on the diffraction data shown partially in Figure 2 in the main text (the irregular large spot in the upper right part of the images comes from shading the radiation by the optical part used for laser heating). We do not consider, however, it would be necessary to include this figure in the manuscript, leaving the final decision to the Editor. The straightforward conclusion that comes into mind is that the main reflections became sharper after the second cycle of laser heating (while the side features turned considerably weaker). While there are no experimental data on temperature dependence of phase V of carbon dioxide, in this study we have confirmed the earlier findings on the negative linear compressibility along the crystallographic c axis. This may suggest that a similarly anisotropic thermal expansion could generate considerable stress during the rapid cooling of the sample to room temperature. The diffraction was taken in the heating spot, since MgCO₃ was distributed quite uniformly in the central part of the pressure chamber.

Line 82: I am not convinced that the diffraction is “less structured” as we still see partial Debye Scherrer rings at 118 GPa.

We have written in the text “less textured”, not “less structured”. Indeed, partial Debye-Scherrer rings are visible still at 118 GPa, but they are more “complete” (hence “less textured”) than at 105 GPa.

Extended Figure 2: Where is the reflection 110 of β -ReO₂? According to Figure 5 of reference 24, I would expect a diffraction peak at roughly $2\theta = 6$.

Indeed, as calculated from the refined lattice parameters the 110 reflection of β -ReO₂ should appear precisely at $2\theta = 6.27^\circ$. It should be noted, however, that as can be seen at the top inset of the Extended Data Figure 2 showing the corresponding 2D diffraction image the diffraction pattern of β -ReO₂ consists of single spots, indicating that we obtained rather clusters of tiny single-crystalline grains than powdered product. Zooming in this inset one can observe single spots scattered at the same 2θ angle inside the first CO₂-V ring, but they are overall too weak and disappear in the background in course of radial integration. To indicate the position of the 110 reflection we have added a tick mark in the Extended Figure 2.

Also, Reference #24 should be added to the figure caption.

We have added the missing reference to the figure caption.

Discussion and Conclusion: In order to meet Nature Communications standards, I think more discussion on the implications of the results is needed. As the authors seem to be willing to discuss their results in terms of Earth Science, I would expect some discussion about the difference in having decomposition of CO₂ with release of oxygen at lower mantle conditions and the stability of a solid

CO₂ phase in terms mantle dynamic and deep carbon cycle (see Isshiki et al 2004; and eventually Hu et al., Nature 2016; Yagi, Nature 2016; Boulard et al. JGR 2012)

We have substantially extended the discussion about the potential implications of our results for Earth science, adding also the suggested references. The contradictory results concerning the decomposition of CO₂ are also discussed in detail in the answer to the first question by reviewer #1.

Methods :

line 296, it is well know that the pressure measured into the Re gasket is significantly lower than the sample at the center of the sample chamber, is that taken into account in the present pressure measurement?

In our experiments we measured the pressure collecting XRD patterns of Re at the edge between the gasket and the high pressure chamber. This methodology is well known and was described in the article by Anzellini et al., J. Appl. Phys. 115 236 , 043511 (2014), cited in the Methods section of our manuscript. According to Anzellini et al., this method of pressure measurement indeed overestimates the pressure by 5% at most (even at pressures above 200 GPa). We therefore estimated the uncertainty of pressure determination as $\pm 5\%$, i.e. about ± 5 GPa at megabar conditions. This uncertainty was included as error bars in Fig. 3 and also regarded while re-plotting the corrected Fig.1. Moreover, our CO₂-V data fits very well with its predicted equation of state (Fig. 3) and the trends in frequencies of Raman modes (Fig. 4), based on the previous experiments by other groups, which confirms that our pressure measurement is correct within the experimental uncertainty.

Extended Figure 3: What are the axis label? Am I understanding correctly that the XRD map was obtain by measuring Re unit cell volume? Does it mean Re diffraction peaks were measurable even in the center of the sample chamber? As Re should only be present in the gasket but not within the sample chamber, I cannot explain to myself why a pressure change would be observed within the sample chamber if only looking at Re XRD peaks.

We agree that this figure was not clear and easy to follow and therefore we decided to remove it from the manuscript.

Reviewer #1 (Remarks to the Author):

This reviewer thanks the authors for the responses to my comments. However, the answers to my points and changes in the article are unsatisfactory. Responding here to each of the four points:

1. The answer is unsatisfactory. Previous experimental work showing decomposition could in fact correspond to higher temperatures; e.g., some degree of molecular dissociation in the fluid to form solid carbon and oxygen. When there is breakdown and heating of different components by different degrees, effects such as Soret diffusion can cause chemical segregation. One can appeal to theory for support but the experimental data need to be addressed on their own merits. See additional points below.

2. I'm pleased to see this point was addressed.

3. I'm pleased to see this point was addressed as well.

4. This point was largely addressed, but the manuscript refers to the amorphous phase and amorphization as if these are thermodynamic concepts. For example, the abstract says 'Even amorphization can be definitely ruled out'. It is sufficient to say that no amorphization is observed.

The above remarks lead to broader points, which mainly concern thermodynamic arguments that are especially problematic in the new version of the paper.

A. It isn't clear what a reaction 'triggered by the Re gasket' means. Either the gasket reacts or it doesn't. In previous experiments, solid oxygen was observed so that oxygen is not incorporated in the gasket. If some sort of 'catalysis' is invoked, the catalyst by definition is neither consumed or produced in the reaction and only *aids* in the system reaching the equilibrium state.

B. The thermodynamic discussion and implications for the lower mantle in the final paragraph is hopelessly confusing. Whether oxygen is released or not in experiments on pure CO₂ provides little information about the multicomponent system corresponding to the minerals and rocks present. A 'mechanism' for diamond formation, for example, is not needed. Kinetic barriers and arguments based on metastability are irrelevant. Over geological timescales, all that matters is the equilibrium state of the chemical assemblage. If the authors were to show that CO₂ exists at the complete P-T-X

conditions of the deep lower mantle, that would be another story. However, they have not reported this, so the closing statement, 'This is supposed to have meaningful implication to the deep carbon cycle and the existing models of the formation of superdeep diamonds', is both strangely worded and unsubstantiated.

C. The extrapolation of the previously reported P-T breakdown line to room temperature and higher pressures is a specious argument. Any number of phase boundaries could intercede before room temperature is reached and there is no reason to believe the previous boundary is linear.

Reviewer #2 (Remarks to the Author):

The authors have addressed most of my comments quite adequately.

However, I find the discussion part still weak as it only addresses what will happen if CO₂ dissociates in the lower mantle and not the becoming of CO₂ if stable. To be short, if CO₂ dissociates at lower mantle conditions: it would produce diamond and fluid oxygen, which in turn significantly affects redox state, increasing oxygen fugacity by several orders of magnitude. The reaction of free oxygen with lower mantle minerals such as Mg-perovskite can create significant conductivity anomalies (Litasov et al., *EPSL*, 2011).

If CO₂ is stable at lower mantle conditions, it could either react with the surrounding oxides (e.g. FeO) to form carbonate and allow transport of carbon down to the CMB (Boulard et al., *PNAS*, 2011 and Boulard et al., *JGR*, 2012); either be reduced in immobile diamonds due to the lower mantle reduced conditions and be at the origin of deep melting upon upwelling of mantle rocks (ie. Redox freezing, Rohrbach et al., *Nature*, 2011).

The answers to Reviewers' comments:

Reviewer #1 (Remarks to the Author):

This reviewer thanks the authors for the responses to my comments. However, the answers to my points and changes in the article are unsatisfactory. Responding here to each of the four points:

1. The answer is unsatisfactory. Previous experimental work showing decomposition could in fact correspond to higher temperatures; e.g., some degree of molecular dissociation in the fluid to form solid carbon and oxygen. When there is breakdown and heating of different components by different degrees, effects such as Soret diffusion can cause chemical segregation. One can appeal to theory for support but the experimental data need to be addressed on their own merits. See additional points below.

2. I'm pleased to see this point was addressed.

3. I'm pleased to see this point was addressed as well.

4. This point was largely addressed, but the manuscript refers to the amorphous phase and amorphization as if these are thermodynamic concepts. For example, the abstract says 'Even amorphization can be definitely ruled out'. It is sufficient to say that no amorphization is observed.

We have removed the questionable sentence from the abstract and added a sentence to the manuscript explaining the metastable character of the amorphous material.

The above remarks lead to broader points, which mainly concern thermodynamic arguments that are especially problematic in the new version of the paper.

A. It isn't clear what a reaction 'triggered by the Re gasket' means. Either the gasket reacts or it doesn't. In previous experiments, solid oxygen was observed so that oxygen is not incorporated in the gasket. If some sort of 'catalysis' is invoked, the catalyst by definition is neither consumed or produced in the reaction and only *aids* in the system reaching the equilibrium state.

It is true that the thorough Raman mapping of our sample after two cycles of laser heating did not reveal any traces of oxygen. On the other hand we have no detailed information on the time of laser heating in the previous studies. Hence, we decide not to speculate on the experimental details of other works leaving the suggestion of possible dissociation spreading from the gasket edge, however without providing thorough explanation of reaction mechanism.

B. The thermodynamic discussion and implications for the lower mantle in the final paragraph is hopelessly confusing. Whether oxygen is released or not in experiments on pure CO₂ provides little information about the multicomponent system corresponding to the minerals and rocks present. A 'mechanism' for diamond formation, for example, is not needed. Kinetic barriers and arguments based on metastability are irrelevant. Over geological timescales, all that matters is the equilibrium state of the chemical assemblage. If the authors were to show that CO₂ exists at the complete P-T-X conditions of the deep lower mantle, that would be another story. However, they have not reported this, so the

closing statement, 'This is supposed to have meaningful implication to the deep carbon cycle and the existing models of the formation of superdeep diamonds', is both strangely worded and unsubstantiated.

We understand the argument that the final paragraph is pretty confusing, we reworded it accordingly with substantial shortening. We removed all the argumentation on the mechanism of diamond formation, which includes the closing statement, and the argument on kinetics and metastability, respectively.

C. The extrapolation of the previously reported P-T breakdown line to room temperature and higher pressures is a specious argument. Any number of phase boundaries could intercede before room temperature is reached and there is no reason to believe the previous boundary is linear.

We thank the reviewer for this suggestion which we added in the new version of the manuscript, providing alternative explanation for the fact that we did not observe any evidence for CO₂ dissociation in our experiments. It is true that one cannot simply extrapolate the existing linear breakdown lines minding the discrepancy between the highest pressure for literature data (~70 GPa) and our experiments (~120 GPa).

Reviewer #2 (Remarks to the Author):

The authors have addressed most of my comments quite adequately.

However, I find the discussion part still weak as it only addresses what will happen if CO₂ dissociates in the lower mantle and not the becoming of CO₂ if stable. To be short, if CO₂ dissociates at lower mantle conditions: it would produce diamond and fluid oxygen, which in turn significantly affects redox state, increasing oxygen fugacity by several orders of magnitude. The reaction of free oxygen with lower mantle minerals such as Mg-perovskite can create significant conductivity anomalies (Litasov et al., EPSL, 2011).

If CO₂ is stable at lower mantle conditions, it could either react with the surrounding oxides (e.g. FeO) to form carbonate and allow transport of carbon down to the CMB (Boulard et al., PNAS, 2011 and Boulard et al., JGR, 2012); either be reduced in immobile diamonds due to the lower mantle reduced conditions and be at the origin of deep melting upon upwelling of mantle rocks (ie. Redox freezing, Rohrbach et al., Nature, 2011).

We thank the reviewer #2 for the added suggestions and the very constructive input related to them. We re-drafted accordingly the closing statement, keeping short and concise (in order to follow the criticism of reviewer #1 related to the final paragraph of the last version).

Reviewer #1 (Remarks to the Author):

The manuscript has been much improved. However, it still has problems.

line 61: remove the word "usually" [i.e., when is the amorphous phase not kinetically trapped?]

line 146 (and below): The authors still seem to think have found they are able to stabilize molecular CO₂ as a pure sample in a high pressure experiment, this means that a) free CO₂ will be present in the lower mantle, and b) this stability in pure form somehow affects geochemical reactions at these depths. The latter is completely wrong.

For example, the conditional, "If CO₂ is stable at lower mantle conditions, it could..." is misleading. "Even if CO₂ is *not* stable at lower mantle conditions, *carbon* could..."

line 144-145: Thus, this should be "evidencing [sic] that CO₂-V can be stable in the absence of chemical reactions with other elements at P-T conditions..."

Reviewer #2 (Remarks to the Author):

The manuscript has been revised well. I recommend that it be accepted for publication.

Reviewer #1 (Remarks to the Author):

The manuscript has been much improved. However, it still has problems.

line 61: remove the word "usually" [i.e., when is the amorphous phase not kinetically trapped?]

We have removed the word "usually" as suggested by the referee.

line 144-145: Thus, this should be "evidencing [sic] that CO₂-V can be stable in the absence of chemical reactions with other elements at P-T conditions..."

We have corrected this sentence according to the referee's comment.

line 146 (and below): The authors still seem to think have found they are able to stabilize molecular CO₂ as a pure sample in a high pressure experiment, this means that a) free CO₂ will be present in the lower mantle, and b) this stability in pure form somehow affects geochemical reactions at these depths. The latter is completely wrong. For example, the conditional, "If CO₂ is stable at lower mantle conditions, it could..." is misleading. "Even if CO₂ is *not* stable at lower mantle conditions, *carbon* could..."

We apologize for having maintained an ambiguous sentence and we absolutely share the referee's objection. We have corrected the statements indicated by the reviewer (lines 146 and 165) removing any possible ambiguity about the existence of "free CO₂" at the P-T conditions relevant to the lower mantle, emphasizing that only the polymeric phase V is stable and can be involved in chemical reactions at these conditions.

Reviewer #2 (Remarks to the Author):

The manuscript has been revised well. I recommend that it be accepted for publication.

We would like to thank the reviewer for the positive feedback.